# Synthesis of a Reactive Template-Induced Core–Shell PZS@ZIF-67 Composite Microspheres and Its Application in Epoxy Composites

**DOI:** 10.3390/polym13162646

**Published:** 2021-08-09

**Authors:** Kunpeng Song, Yinjie Wang, Fang Ruan, Weiwei Yang, Zhuqing Fang, Dongsen Zheng, Xueli Li, Nianhua Li, Meizhuang Qiao, Jiping Liu

**Affiliations:** 1School of Materials Science and Engineering, Beijing Institute of Technology, 5 Zhongguancun South Street, Haidian District, Beijing 100081, China; 13439970074@163.com (K.S.); 18255171997@163.com (F.R.); yangweiwei0811@163.com (W.Y.); 15733185216@163.com (X.L.); 1788819050@163.com (N.L.); qao_mz@163.com (M.Q.); 2School of Mechatronical Engineering, Beijing Institute of Technology, 5 Zhongguancun South Street, Haidian District, Beijing 100081, China; 3220195023@bit.edu.cn (Z.F.); 3120195165@bit.edu.cn (D.Z.)

**Keywords:** flame retardant, polyphosphazenes, ZIF-67, epoxy resin, mechanism of flame retardant

## Abstract

Developing superior properties of epoxy resin composites with high fire resistance, light smoke, and low toxicity has been the focus of the research in the flame-retardant field. In particular, it is essential to decrease the emissions of toxic gases and smoke particles generated during the thermal decomposition of epoxy resin (EP) to satisfy the industrial requirements for environmental protection and safety. Consequently, the PZS@ZIF-67 composite was designed and synthesized by employing the hydroxyl group-containing polyphosphazene (poly(cyclotriphosphazene-co-4,4′-dihydroxydiphenylsulfone), PZS) as both the interfacial compatibility and an in situ template and the ZIF-67 nanocrystal as a nanoscale coating and flame-retardant cooperative. ZIF-67 nanocrystal with multidimensional nanostructures was uniformly wrapped on the surface of PZS microspheres. Subsequently, the acquired PZS@ZIF-67 composite was incorporated into the epoxy resin to prepare composite samples for the study of their fire safety, toxicity suppression, and mechanical performance. Herein, the EP/5% PZS@ZIF-67 passed the V-0 rating in a UL-94 test with a 31.9% limit oxygen index value. More precisely, it is endowed with a decline of 51.08%, 28.26%, and 37.87% of the peak heat release rate, the total heat release, and the total smoke production, respectively. In addition, the unique structure of PZS@ZIF-67 microsphere presented a slight impact on the mechanical properties of EP composites at low loading. The PZS@ZIF-67 possible flame-retardant mechanism was speculated based on the analysis of the condensed phase and the gas phase of EP composites.

## 1. Introduction

Epoxy resin (EP) has been widely used as an advanced thermosetting polymer with excellent mechanical properties, electrical insulation properties, adhesive properties, and flexibility in the manufacturing industry’s use process [1,2,3,4,5]. However, the poor fire resistance and the severe release of smoke and toxic gas during the EP combustion severely limits its further application potential in the high-performance composite matrix fields [6,7]. Therefore, flame retardants are employed to perform flame-retardant treatment on polymer materials to avoid material combustion and fire spread, contributing to the polymer composite exhibiting smoke suppression, self-extinguishing, and flame retardants [8]. Over the past decades, various phosphorus- [9], nitrogen- [10], sulfur- [11,12], and silicon-containing [13,14,15] compounds have been incorporated into the epoxy matrix to strengthen the fire resistance properties. Moreover, strategies of doing multiple flame-retardant elements and designing multidimensional geometric structures have also been reported to improve composites’ comprehensive properties [16,17]. Polyphosphazene is a multifunctional organic–inorganic hybrid material composed of alternating phosphazene units (-P=N-), and its active side groups contribute to the diversity and flexibility of phosphazene materials [18,19,20]. Since the successful synthesis of nano-and microscale polyphosphazene several years ago, such as polyphosphazene microspheres, polyphosphazene nanotubes, or nanofibers, organic–inorganic hybrid materials have fascinated researchers from multiple fields, such as biomaterials, electrical materials, optical materials, etc. [21,22,23]. During this decade, a series of polyphosphazene derivatives have been reported and employed as crosslinking agents, reinforcing agents, and flame-retardant additives to optimize composites’ overall performance. Ma reduced the chloroplatinic acid on PPS microspheres’ surface to synthesize platinum nanoparticle-containing polyphosphazene submicrospheres (Pt-PPS). Extremely low Pt content significantly improved the flame retardants of epoxy composites. The catalytic carbonization of Pt and PPS sub-microspheres contributed to the improvement of flame retardants [24]. Zhu assembled poly (cyclotriphosphazene-co-4,4′-sulfonyl diphonel) nanotubes onto a carbon fiber surface by in situ template polymerization as a new type of multi-scale hybrid reinforcement material. In addition, this modification enhanced the flame retardants of epoxy composites and even ameliorated their mechanical properties [25]. Xu reported that poly (cyclotriphosphazene-co-4,4′-sulfonyl diphenol) nanotubes decorated with Ag nanoparticles have excellent thermal stability. Designed composites presented perfect catalytic activity and reusability [26]. Multiple functionalizations such as special structure composition and heteroatoms doping can endow these materials with surface modifiability, chemical stability, electrical conductivity, catalytic performance, etc.

Novel hybrid nanomaterials with particular surface chemistry and multidimensional geometric nanoarchitecture may promote advanced materials’ rapid evolution. As an important branch of MOFs (metal–organic frameworks), zeolite-like metal–organic frameworks (ZIFs) integrate the advantages of zeolite and MOFs, such as high porosity, cage-like cavities, adjustable pores, designate functions, excellent stability, etc., which caused great application prospects in the fields of efficient catalysis [27,28], energy storage [29,30], gas adsorption separation [31,32], sensor detection [33,34], and immobilized enzymes [35,36]. In recent years, researchers have been focusing on regulating MOFs’ morphology and structure, especially making MOFs exist as ultra-thin flakes, which can effectively increase the coordinatively unsaturated metal sites and further improve their application performance. The core–shell structure is a layered material structure that makes up for each other’s deficiencies to some extent and also integrates the advantages of the inner and outer layers of materials [37]. It is the preferred strategy for researchers to design and synthesize novel functional materials in recent years. Due to the crystal’s anisotropy, the construction of core–shell MOFs often presented differences in shell thickness in different directions due to different growth rates on diverse crystal planes. In response, Gong provided a novel strategy for constructing core–shell MOFs, which is to encapsulate different guest molecules in layers in the same host framework. This work provides new ideas for the bottom–up construction of core–shell MOFs materials and establishes a basis for the further evolution of crystalline multifunctional host–guest systems [38].

In this study, ZIF-67 nanocrystals were uniformly anchored on the surface of PZS microspheres taking advantage of the active nature of PZS surface and the characteristics of ZIFs to enhance interfacial interaction. The thermal stability of the flame retardant was improved by coating ZIF-67 nanocrystals on PZS. The presence of phosphorus, nitrogen, sulfur, and cobalt elements endows the composite multiple synergistic flames retarding effect and facilitates the formation of a solid char layer during the composites’ combustion. Thanks to this novel flame retardant PZS@ZIF-67, the thermal properties and flame retardancy of epoxy composites were significantly enhanced while only slightly affecting on its mechanical properties.

## 2. Materials and Methods

### 2.1. Materials

2-methylimidazole (2-MeIM) (≥98%) was supplied by the Tianjin Komiou Chemical Reagent Co., Ltd. (Tianjin, China). Cobalt nitrate hexahydrate (Co(NO_3_)_2_·6H_2_O) (≥98%) was obtained from Macklin Biochemical Co., Ltd. (Shanghai, China). Hexachlorocyclotriphosphazene (HCCP) (≥99%) was purchased from Zibo Lanyin Chemical Co., Ltd. (Sandong, China). 4,4-dihydroxydiphenylsulfone (BPS) (≥99%) was provided by Aladdin Chemical Reagent Co., Ltd. (Shanghai, China). Trimethylamine (AR), acetonitrile (AR), and acetone (AR) were obtained from the Weiss Chemical Reagent Co., Ltd. (Beijing, China). Diglycidyl ether of bisphenol A (E-44) was provided by the Nantong Xingchen Synthetic Material Co., Ltd. (Hunan, China). Acetonitrile, absolute methanol, and 4,4-diaminodiphenylmethane (DDM, ≥98%) were obtained from the Beijing Chemical Plant (Beijing, China). Acetonitrile was dried over 4 Å molecular sieves before use. 

### 2.2. Preparation of PZS@ZIF-67

The schematic illustration of the preparation of PZS@ZIF-67 was shown in Figure 1. PZS were obtained according to the previously reported method [39]. The preparation of core–shell PZS@ZIF67 follows a typical run. Firstly, a certain amount of PZS was dispersed into 50 mL of absolute methanol with the assistance of ultrasonication for 1 h, which was followed by the dissolving of Co(NO_3_)_2_·6H_2_O (0.3 mmol), which was labeled as solution A. Then, 2-MeIM (1.2 mmol) and TEA (1.2 mmol) were dissolved into 50 mL of absolute methanol to form a clarified solution B. Subsequently, solution B was poured into solution A with a quick stirring for several hours. The product was collected by centrifugation, washed several times with absolute methanol, and dried at 80 °C for 10 h under the vacuum oven. Moreover, ZIF-67 was prepared by following the above operating steps without adding PZS.

### 2.3. Preparation of Epoxy Composites

Firstly, different mass fractions of PZS@ZIF-67 (1, 3, and 5 wt %) were dispersed in a certain amount of acetone with the assistance of ultrasonication for 30 min, then spread into the epoxy resin under stirring at 80 °C, until the acetone is completely evaporated. Subsequently, the hardener DDM (ratio of DDM/epoxy resin was 1:4) was coupled with the above solution. The mixture was stirred at 80 °C for 1 min to form a homogeneous liquid, which was placed into the vacuum oven at 80 °C for 3 min to eliminate bubbles and immediately poured into the pre-heated molds of specific sizes. Finally, the mixtures were cured in the blast oven at 120 °C for 2 h and then heated to 150 °C for 4 h to obtain EP composites. The prepared samples were denoted as EP/1% PZS@ZIF-67, EP/3% PZS@ZIF-67, and EP/5% PZS@ZIF-67. The schematic diagram of the preparation method is shown in Figure 2.

### 2.4. Characterization

Fourier transform infrared (FTIR) spectroscopy was performed on a Tensor 27 IR spectrometer (BRUKER OPTICS, Beijing, China). Spectra were collected at 32 scans with a spectral resolution of 4 cm^−1^, and the test range is 500–4000 cm^−1^. X-ray powder diffraction (XRD) patterns of the samples were recorded using Cu-Kα radiation at 40 kV and 15 mA (2θ ranges from 2° to 10° with a step size of 0.02°). Thermogravimetric analysis (TGA) and Differential scanning calorimetry (DSC) were performed with a TOLEDO STARE thermal analyzer (Mettler-Toledo, Zurich, Switzerland); measurements were carried out in nitrogen atmosphere from 50 to 800 °C with a heating rate of 20 °C min and 50 to 200 °C with a heating rate of 5 °C/min, respectively. Limiting oxygen index (LOI) values were recorded with a FTAII 1600 LOI instrument (Rheometric Scientific Ltd., Suzhou, China) using the standard ASTM D 2863 procedure; the sample dimensions were 130 × 6.5 × 3 mm^3^. Vertical burning tests were carried out using a CZF-5 horizontal vertical burning tester (Phoenix Instruments Co., Ltd., Suzhou, China) with the UL-94 standard; the samples of dimensions were 130 × 13 × 3 mm^3^. Cone calorimeter measurements were performed with Fire Testing Technology (FTT) apparatus (Phoenix Instruments Co., Ltd., Suzhou, China) with a truncated cone-shaped radiator and the measurements according to the ISO 5660 protocol. The specimen (100 × 100 × 3 mm^3^) was measured horizontally without any grids. The results were averaged from two measurements. Scanning electron microscopy (SEM) images were obtained with a Hitachi SU8020 (Hitachi Limited, Tokyo, Japan) with a 15 kV accelerating voltage. Specimens were prepared by placing a drop of the sample dispersed in acetone on an aluminum foil, and the specimens were dried in air. Specimens were sprayed with a thin gold layer to make good electrical surface conductivity. Transmission electron microscope (TEM) images were taken using a transmission electron microscope (HT7700, Hitachi, Ltd., Tokyo, Japan) with a 120 kV electron source. Specimens were prepared by placing a drop of the sample in acetone on a carbon-coated copper grid, and the specimens were dried in air. X-ray photoelectron spectroscopy (XPS) patterns of the samples were carried out with Quantera II X-ray photoelectron spectroscopy (Ulvac-PHI, Chigasaki, Japan), and the measurements were performed at 25 W power and 10^−6^ Pa vacuum. The mechanical properties of splines were tested by the general testing machine (DXLL-5000, Shanghai, China) in accordance with GB/T 2567-2008 “Tensile Test and Bending Test Method”. Thermogravimetric analysis (Mettler-Toledo, Zurich, Switzerland) was coupled with Fourier transform infrared spectroscopy (BRUKER OPTICS, Beijing, China), which was performed under nitrogen atmosphere from 50 to 800 °C at the heating rate of 20 °C/min^−1^. 

## 3. Results

### 3.1. Structural Characterization of PZS@ZIF-67 Composites

After studying the preparation process of the sample, SEM images of PZS and PZS@ZIF-67 were obtained and are shown in Figure 3a,c respectively. PZS microspheres are symmetrical sphericity with an average size of 2 um. Notably, a layer of well-distributed ZIF-67 nanocrystals was wrapped on the exterior surface of PZS. Obviously, compared to the original PZS microspheres, the composite microspheres have a rougher surface and a thicker diameter. The PZS@ZIF-67 showing a core–shell structure was synthesized by in situ self-assembly of ZIF-67 nanocrystals on the PZS microspheres surface using it as a template. 

To further confirm the structure of PZS@ZIF-67, TEM images of PZS and PZS@ZIF-67 were taken and are given in Figure 3b,d respectively. After Co(NO_3_)_2_·6H_2_O and 2-MeIM self-assembly process on the PZS surface, a uniformly dispersed layer of the organic frame with the thickness of 20–50 nm is tightly coated onto the external surface of PZS. Notably, ZIF-67 is evenly distributed on the surface of PZS.

FTIR characterization was carried out to confirm the reaction and composition of the prepared samples, and the results are shown in Figure 4a. The typical peaks of the HCCP spectrum at 1162, 871, and 575 cm^−1^ correspond to P=N, P-N, and P-Cl stretching vibrations [40]. The characteristic absorption peak of the BPS spectrum at 3420 cm^−1^ is assigned to -OH stretching vibration, and the apparent peaks at 1594 and 1491 cm^−1^ are attributed to the stretching vibration of the C=C group in the benzene ring units, respectively [41]. From the PZS spectrum, it could be observed that two absorption peaks of P-N and P=N are retained, and the characteristic peak intensity of the P-Cl is very low. In addition, distinguishable peaks of benzene rings appeared at 1594 and 1491 cm^−1^, indicating that PZS was successfully synthesized. After coating with ZIF-67, a broadband located at 3502 cm^−1^ belongs to the unreacted imino of 2-methylimidazole and residual trimethylamine in ZIF-67 mesoporous, and the characteristic peak at 759 cm^−1^ corresponds to the bending vibration peak of the imino. In addition, the PZS@ZIF-67 spectrum retains the basic structure of PZS, and the absorption peaks of ZIF-67 occur in the low wavenumber range of 500–800 cm^−1^ that corresponds to the vibration of the NO_3_^−^ group and the Co−N lattice vibration [42,43,44], indirectly revealing that ZIF-67 thinly coated PZS. 

XRD examined the phase and composition of PZS@ZIF-67. The magnified pattern (Figure 4b) indicates reflections from the lattice planes, 2θ = 7.4°, 10.4, 14.8°, 16.5°, 18.1°, and 26.8° typical of ZIF-67. PZS shows a batch of miscellaneous amorphous peaks, which match well with the previous report [29]. Notably, after coating with ZIF-67, some new broadbands at 7.4°, 18.1°, and 26.8° emerge for composite PZS@ZIF-67. Results showed that part of the characteristic peaks belonging to ZIF-67 could be discovered in PZS@ZIF-67. The above evidence proves that the PZS and ZIF-67 were successfully combined to form the composite. Co^2+^ and 2-MeIM obtained by deprotonation could not coordinate regularly well under the complex reaction environment, and the final obtained ZIF-67 was not the customary rhomboid hexahedron structure. Only part of the crystal plane information of ZIF-67 could be obtained, which is in agreement with the above TEM results.

The elemental composition of PZS@ZIF-67 was identified from XPS spectra. Figure 4c showed XPS spectra of PZS, and the existence of C, N, O, P and S have also been confirmed. After wrapping with ZIF-67, the XPS spectral line of PZS@ZIF-67 displayed the new band of Co element. Notably, the N, P, and S are retained due to the nanoscale effect of ZIF-67 and the incomplete coating effect. XPS results perfectly demonstrate that Co, N, and C elements belonging to ZIF-67 are wrapped on the surface of PZS. 

TGA studied the thermal performances of PZS, ZIF-67, and PZS@ZIF-67 in a N_2_ atmosphere, and the results are shown in Figure 4d and Table 1. The onset decomposition temperature (the temperature at 5% mass loss, T_5%_) of ZIF-67 reaches 364 °C, and the maximum decomposition temperature (the temperature at the maximum mass loss, T_max_) is over 510 °C, with 34.8 wt % residual yields at 800 °C, indicating that the ZIF-67 possesses distinguished thermal stability. PZS displays a two-step thermodynamic decomposition behavior. The initial mass loss stage of PZS occurs between 50 and 200 °C, corresponding to the removal of free water and the dehydration reaction between the active hydroxyl in the PZS structure. The second mass loss behavior between 450 and 600 °C is due to rigid structure decomposition of the phosphazene ring and the aromatic components of the BPS. PZS@ZIF-67 presents a two-stage degradation process, exhibiting similar decomposition behaviors compared with PZS. Notably, PZS@ZIF-67 preserves a better thermal performance accompanied by a higher T_5%_ at 483 °C and a lower quality loss, which may be due to the binding effect between PZS surface and ZIF-67. Additionally, transition metal Co^2+^ acts as a catalyst for the crosslinking of polymers, resulting in increased residual mass.

### 3.2. Dispersion of PZS@ZIF-67 in Epoxy Composites

To further observe the state of dispersion of the flame-retardant additives, the morphology and microcosmic structures were characterized by SEM and EDX, as shown in Figure 5. To a large extent, the volume and structure of the filler in the boundary sandwich determine the properties of the composite [45]. It is observed that PZS in the epoxy polymer has a particle-state morphology, and no significant aggregation was observed (Figure 5c,d). Moreover, there is an apparent interface between PZS and epoxy matrix, which hurts the mechanical properties of the composites by causing stress concentration under external loads. As showed in Figure 5e,f, PZS@ZIF-67 particles are more well dispersed than PZS particles. Interestingly, the PZS@ZIF-67 interface is well bonded to the epoxy substrate, which is most likely due to the coating’s nanoscale effect. Thus, using the nano-coating method introduced in this work, more uniform dispersion and better interfacial compatibility for the modified PZS microspheres were achieved in the epoxy matrix, which was found to have an advantageous effect on various performances.

### 3.3. Thermal Stability of Epoxy Composites

The thermal performances of pristine EP and its composites were studied by TGA under nitrogen atmosphere (Figure 6a,b). All the samples presented the one-stage thermal decomposition process, and the EP/PZS@ZIF-67 composites showed thermal decomposition behaviors similar to that of the EP/PZS composites. The incorporation of fillers reduces the composites’ thermal stability compared to that of pristine EP, accompanying decreased T_5%_ values, which show that the catalytic effect of fillers is inseparable from the accelerated thermal decomposition behavior of EP composites. More accurately, the appropriately earlier preliminary decomposition of the epoxy composites facilitates the char layer’s formation, which could effectively inhibit the fire spread. In addition, the thermal stability of the EP/PZS@ZIF-67 composite is higher in terms of increased T_5%_ and T_max_ (Table 2) compared to that of the EP/PZS composite, which is attributed to coating with thermally stable ZIF-67 nanocrystalline on the surface of PZS, preventing PZS from catalyzing the epoxy matrix decomposition in advance. Interestingly, the decomposition interval of epoxy composites is generally wider, and the peak value is smaller with the addition of PZS@ZIF-67, indicating that PZS@ZIF-67 microspheres can reduce the epoxy matrix decomposition rate and delay the combustion of the epoxy substrate. Additionally, it is easy to observe that EP/PZS@ZIF-67 composites exhibit better thermal stability than EP/PZS composites, which is mainly due to the synergistic effect of transition metals and phosphazene for the acceleration of decomposed macromolecular chains being crosslinked. The increased residual mass is able to play a physical shielding role, inhibiting the transmission of combustible gas between the atmosphere and the EP matrix. However, under the same load, the residual mass percentage at 600 °C and 800 °C of the EP/PZS@ZIF-67 composites is slightly lower than the EP/PZS composites, which is possibly due to the introduction of ZIF-67 slightly affecting the char formation mechanism of PZS on the epoxy matrix.

Studying the glass transition temperature (T_g_) of the epoxy and its composites is a critical evaluation criterion for high-performance epoxy composites, and detailed information is presented in Figure 6c. All epoxy composites present only one T_g_, indicating that both PZS@ZIF-67 and PZS have good compatibility with the epoxy matrix. Compared with pristine EP, the T_g_ of containing PZS@ZIF-67 composites gradually increased with the increase of PZS@ZIF-67 content. However, the T_g_ of EP/PZS@ZIF-67 composites is slightly higher than that of EP/PZS composites at the same filler content. The increase in the T_g_ of epoxy composites may be attributed to the active hydroxyl groups on the flame-retardant filler’s surface and the ZIF-67 nanocrystals with excellent compatibility, which leads to crosslinking of the filler and epoxy monomers during the curing process.

The curing behavior of epoxy monomers under different systems was tested to explore further the influence of fillers in the curing process (Table 3). As shown in Figure 6d, pristine EP displays a prominent exothermic peak near 142 °C, which is caused by the breakage and recombination of the chemical bond between DDM and epoxy resin in the crosslinking reaction. Compared with pristine EP, the two EP composites (EP/5% PZS and EP/5% PZS@ZIF-67) also exhibit similar curing behavior, and the exothermic peaks are both advanced. This is probably a weak interaction between the filler and the epoxy system, which shortens the distance between the DDM and the epoxy monomers. However, for the mixed system of epoxy monomers and fillers without hardener DDM (EP-5% PZS and EP-5% PZS@ZIF-67), almost no exothermic peak is observed near 140 °C. Therefore, it can be preliminarily proved that the filler will not have an obvious chemical reaction with DDM during the epoxy curing process. It is preliminarily inferred that the T_g_ of epoxy composites’ improvement is inseparable from the weak interaction between the filler and the epoxy matrix, such as hydrogen bonds and van der Waals forces.

### 3.4. Fire Safety of Epoxy Composites

The fire behavior of the pristine EP and its composites was initially investigated through LOI and UL-94 vertical burning tests to examine the flame-retardant properties, and the results are listed in Table 4. Pristine epoxy presents an LOI value of 25.5%, and no rating in the UL-94 vertical burning test with severe melt dripping. We can find out that the increase of PZS@ZIF-67 content in epoxy composites from 0% to 5%, this LOI value gradually increases from 25.5% to 31.9%. Notably, PZS@ZIF-67 tends to be a better flame-retardant for epoxy matrices than PZS under the uniform load in composites. The incorporation of 3% PZS did not make epoxy composites reach any rating in the UL-94 test, while epoxy composites containing 3% PZS@ZIF-67 achieved a V-1 rating without dripping. This suggests that the combination of ZIF-67 and PZS has a positive effect on the flame retardant composite. The improvement of the flame retardants of epoxy composites may be attributed to the following two aspects. On the one hand, matching organic flame retardant elements with transition metal can better play the role of catalytic char formation in the condensed phase. On the other hand, ZIF-67 nanocrystals grown on PZS could effectively reduce the aggregation of ZIF-67 to achieve nanoscale effects better. Since epoxy composites under 5% flame retardant additive load exhibited satisfactory LOI and UL-94 tests, the cone calorimeter test was further performed on the above epoxy composites to study their fire resistance.

The cone calorimeter test was carried out on these samples to further study their combustion behaviors under real-world fire conditions. The heat release rate (HRR), total heat release (THR), and total smoke production (TSP) vs. time curves for epoxy composites were obtained in Figure 7. Furthermore, several key parameters, such as the peak heat release rate (pHRR), THR, TSP, the fire growth rate index (FIGRA), and residual mass percentage obtained from cone calorimeter are listed in Table 5.

Pristine EP is extremely flammable, showing a sharp peak of heat release rate with a peak value of 1156.16 kw/m^2^. Compared to pristine EP, the pHRR values of EP/5% PZS and EP/5% PZS@ZIF-67 composites are decreased to 640.69 and 565.59 kW/m^2^, corresponding to 44.58% and 51.08% reduction, respectively. 

For 200 s, pristine EP presents the total heat release value of 78.16 MJ/m^2^, and the introduction of fillers significantly reduced the value. Moreover, the THR of EP/5% PZS@ZIF-67 decreases more obviously and grows more slowly than that of EP/5% PZS composite, presenting only a total heat release value of 56.07 MJ/m^2^ at 200 s. This lower release value indicates that more epoxy polymers are maintained in the condensed phase, and fewer volatiles are released into the gas phase during combustion, which may be due to the good synergistic effect between cobalt ion and phosphorus–nitrogen flame-retardant elements for catalyzing epoxy molecular chain into charring. 

In addition to the combustion heat hazards, smoke release hazards are also essential criteria for flame-retardant polymers to be taken into account. Consistent with previous research reports, phosphorus-containing flame retardants exhibit a high flame-retardant efficiency for EP composites but have no significant effect on reducing smoke emission. Of all the epoxy composites, EP/5% PZS@ZIF-67 composites present the lowest TSP value of 16.16 m^2^, with a 37.80% reduction compared to pristine epoxy resin, and the incorporation of flame-retardant fillers exhibits an increasing inhibition effect as the filler content increased. The influence of PZS@ZIF-67 on the TSP appears in a similar trend to that of the THR. Reduced organic volatiles content inhibits combustibles’ thermal feedback in the epoxy resin and reduces the primary source of smoke particles. In addition, the increased residual mass percentage also well supports the catalytic effect of the PZS@ZIF-67 composites (Table 5). 

The fire growth rate index (FIGRA), the maximum ratio of the sample’s heat release rate to its corresponding time during the combustion process, which is usually equivalent to the ratio of PHRR/time to pHRR, is another crucial derivative parameter related to combustion performance classification. The FIGRA value of flame-retardant composites was observably reduced compared with neat epoxy resin (13.72 kW/m^2^s). It is worth noting that the more significantly reduced FIGRA value is observed for the EP/PZS@ZIF-67 composite than those EP/PZS composites, showing the higher fire safety performance among the same loading samples.

### 3.5. Condensed Phase Analysis

Regarding the condensed phase flame-retardant mechanism, char residues of the epoxy composites after cone calorimeter tests are shown in Figure 8. Figure 8a,d shows macro-morphologies of selected samples’ char residues by using a digital camera. Observing from different perspectives, it is not difficult to find that the neat EP was almost completely burned, while the other two samples have much higher char yields. At first sight from the front view in Figure 8e,f, the char layer structure of EP/5% PZS presented to be more intumescent than that of EP/5% PZS@ZIF-67. However, the char structure of EP/5% PZS@ZIF-67 was more compact without significant fracture in favor of serving as barrier layers.

The microstructures of residual chars were taken with SEM and are shown in Figure 9a–f. The EP is completely burned out, which presents serious fracture under the microscopic. The residual char structure of two flame-retarded samples is very similar. However, the exterior and interior char layers of EP/5% PZS@ZIF-67 are more compact and continuous than those of EP/5% PZS, which implies that it had more robust char layers. In particular, a large number of small ruptured holes were formed in the interior char layers of two flame-retarded samples after burning, which is consistent with its HRR curve and TSR curve, indicating that heat and smoke particles slowly escaped through these channels.

The exterior and interior residual chars of EP, EP containing 5% PZS, and EP containing 5% PZS@ZIF-67 were studied with XPS, and the test data were collected and summarized in Table 6. Figure 10a presented full survey XPS spectra of residual chars, and the existence of C, N, O, P, and Co were also confirmed. The C-C and C-H in aliphatic and aromatic species, C-O (ether, ester, and hydroxyl group) and C=O, could be observed from the characteristic C1s bands to 284.6, 286.8, and 288.4 eV, respectively. Characteristic Co2p bands at 785.2, 801.8 eV, and 782.9, 800.0 eV are ascribed to Co^2+^ and Co^3+^ (Figure 10h,k), implying that cobalt exists in the residual char in the form of divalent and trivalent oxidation states. Moreover, satellite peaks at 789.2 eV and 805.6 eV correspond to Co2p3/2 and Co2p1/2, respectively. In addition, the cobalt, phosphorus, and nitrogen content in the exterior residual char is significantly higher than that in the interior char, which is inseparable from the PZS@ZIF-67 to migrate to the exterior of the epoxy composites. Apart from this, XPS results perfectly demonstrate that the accumulation of an inorganic layer composed of transition metal oxides and phosphorus–nitrogen compounds on the char surface improves the residual chars’ thermal oxidation resistance, effectively inhibiting the penetration of oxygen and the release of combustible gases.

### 3.6. Vapor Phase Analysis

TG-FTIR was utilized to analyze the gaseous pyrolysis products generated during the thermal decomposition process. As shown in Figure 11, FTIR spectra were obtained by analyzing the evolution rate of gaseous products during the EP’s thermal decomposition and its composites at different characteristic temperatures. The special structural composition of EP composites makes it generate several small molecular gases during the decomposition process, which are accompanied by strong infrared absorption signals, such as -C-H groups for acetone, ether/ester compounds (1178 and 1257 cm^−1^), various hydrocarbons (2800–3100 cm^−1^), water (3650–3735 cm^−1^), ammonia (3340–3540 cm^−1^), aldehyde/ketone (1740 cm^−1^), and aromatic compounds (1605, 1510, and 1460 cm^−1^).

At T_5%_, EP thermodynamically decomposes to release pyrolysis products, presenting a series of obvious infrared absorption peaks. The absorption intensity of EP/5% PZS was lower than that of EP but higher than that of EP/5% PZS@ZIF-67, suggesting that the initial decomposition process is relatively slow for EP composites. Next, EP and its composites both show obvious gas-phase product absorption peaks at T_max_, and peak positions are virtually the same, indicating that pyrolysis products are highly similar. As the temperature rises further, the intensity of aromatic compounds increases for EP composites but is still lower than that of neat EP. Notably, the fingerprint region of 650–910 cm^−1^ is also called the benzene ring substitution region, which implies that benzene rings’ different substitutions will be reflected in this range. Introducing flame-retardant fillers slightly changed the composition of aromatic compounds at the decomposition products of the epoxy matrix, from mono- (751 and 680 cm^−1^) and tri-substituted (831 and 910 cm^−1^) aromatic compounds to di- (751 cm^−1^) and tri-substituted (680 and 831 cm^−1^) compounds.

In short, the spectra of the three EP samples was significantly different from each other. However, the FTIR spectral characteristic peak intensity of gaseous products in the spectrum of EP/5% PZS@ZIF-67 was much lower than that of the other two samples. Lower absorption intensity for gas phase pyrolysis products is mainly attributed to the flame-retardant additives in catalyzing the epoxy matrix to form an inflated and compact char structure, which severed as a consolidated barrier layer, preventing the pyrolysis products from escaping to the outside.

The source of smoke is mainly assigned to the particles generated by insufficient combustion of hydrocarbons and aromatic organic volatiles. It is acknowledged that smoke and toxic gas production is decreased during the combustion process, which endows the EP composite with more reliable fire safety. Next, it was further studied that the representative pyrolysis products’ absorption intensity of three EP samples changes over time (Figure 12).

Compared with pure EP, the release of methane, hydrocarbons, aromatic compounds, and ether compounds of EP/5% PZS and EP/5% PZS@ZIF-67 are inhibited, and EP/5% PZS@ZIF-67 are lower than that of EP/5% PZS, which is inextricably related to the cooperative effect of ZIF-67 and PZS. Moreover, the decomposition behavior of samples modified by flame retardant was advanced, which was accompanied by a narrower high-intensity gas release range segment, which is distinctly different from pure EP. The evolutionary behavior of water and ammonia in the flame-retardant composites is very similar, but both of them are higher than that of pure EP, which is caused by the higher dehydration char formation mechanism and conversion rate in the epoxy crosslinked structure. The significant decrease of carbon-containing gas emissions is mainly ascribed to the catalytic effect of flame-retardant additives on carbon production of condensed phase. The formation of an isolated char layer effectively blocks the flow of smoke particles to the gas phase.

### 3.7. Mechanical Properties of Epoxy Composites

Figure 13a,b displays the tensile and flexural strength for neat EP and its composites. The incorporation of PZS@ZIF-67 slightly improved both tensile and flexural strength than those of EP/PZS, but those values were higher than pristine EP under a relatively low loading. Multiplex factors may contribute to the better tensile and flexural strength in PZS@ZIF-67-added composites compared with PZS-added composites. One noticeable factor is the optimized dispersion of the PZS@ZIF-67 in the epoxy matrix because of the in situ self-assembly method of ZIF-67 nanocrystalline on the surface of PZS. Another reason is ascribed to the enhanced interfacial interaction between the flame-retardant additive and the matrix that facilitated stress dispersion.

## 4. Discussion

The flame-retardant mechanism of PZS@ZIF-67 was speculated based on the investigation of residual chars and pyrolysis products (Figure 14). Neat EP produces many pyrolysis gases during the combustion process, including nitrogen oxides, aromatic compounds, carbonyl compounds, carbon oxides, esters and ethers, and hydrocarbons, which are considered to have the smoke poisoning hazard. Through the synergy of the catalytic action of ZIF-67 and the gas-phase radical quenching effect of PZS, the carbon oxides, hydrocarbons, and other pyrolysis gases are released into the gas phase during the decomposition of EP composites and transformed to the condensed phase, prompting the formation of a highly graphitized char layers. The residual char is captured in the previous polymer combustion stage and subsequently catalytically forms the dense coke layer to slow down the further decomposition of the matrix, which also inhibits the release of smoking production. The transition metal cobalt ion in ZIF-67 catalyzed the polycyclic aromatic reaction generated from the epoxy matrix pyrolysis to form the high-quality carbonaceous components attached to the organic–inorganic hybrids. The phosphorus–nitrogen–cobalt-based catalyst is evenly distributed between the PZS@ZIF-67, and the epoxy matrix catalyzes the surrounding region to form solid char sheets, which are crosslinked together to form continuous and dense char layers. The formed residual charcoal, which presents more graphitized components and higher mechanical strength, interacts with the volatiles produced by degradation to form the intumescent char layers. PZS@ZIF-67 composites are converted into crosslinked phosphate compounds, cobalt metal oxides, cobalt metal phosphates and carbonized aromatic networks during complex combustion processes [46]. In addition, the phosphorus and nitrogen compounds and carbonized aromatic networks formed during the EP/PZS@ZIF-67 combustion process endow the residual char with higher thermal stability and compactness. EP/PZS@ZIF-67 composites show the higher graphitized and more residual char, which is considered an ideal protective barrier layer to effectively suppress the gas and thermal energy exchange between the interior matrix and exterior environment [47,48]. The designed PZS@ZIF-67 core–shell structure presents a large specific surface area, which may increase the interaction area with pyrolysis products that promote physical and chemical adsorption and catalytic reactions. Moreover, good interfacial compatibility between PZS@ZIF-67 and epoxy matrix played a significant role in the cooperation effect between PZS and ZIF-67 nanocrystals.

Apart from the excellent chemical characteristics of PZS@ZIF-67, its outstanding physical properties play a positive role in enhancing the fire safety performance of the as-prepared sample. Owing to the proposed “in situ self-assembly manufacturing method”, ZIF-67 nanocrystals were successfully grown on the surface of PZS and acted as “isolating agents” to inhibit the microspheres from agglomerating into larger particles, which is beneficial to heightening the filler dispersibility in the epoxy matrix. Additionally, when compared with unmodified PZS microspheres, we designed a core–shell structure to improve the PZS surface’s roughness, which strengthens the interface interaction with the epoxy matrix. The modification of PZS by ZIF-67 nanocrystalline ameliorates the interfacial compatibility between the matrix and the filler to a certain degree, which has a positive advantage in the collaborative effect of phosphorus–nitrogen in PZS and catalytic function of ZIF-67. 

The synergy between PZS and ZIF-67 was closely correlated to the catalytic reaction between phosphoric acid and metaphosphoric acid in OP^+^, O_2_P^+^, and HO_2_P^+^ formed PZS decomposition and Co_2_O_3_ formed during the ZIF-67 nanocrystalline decomposition, which improved the mechanical strength of the formed char barrier. Therefore, these advantages of the PZS@ZIF-67 effectively inhibit the emission of smoke particles and toxic gas and reduce HRR and THR, which fully shows that EP/PZS@ZIF-67 composites’ safety has been greatly improved.

## 5. Conclusions

In this work, core–shell PZS@ZIF-67 derived from ZIF-67 nanocrystalline were uniformly amended on PZS microspheres’ surface through a “self-assembly method” to prepare the PZS@ZIF-67 composite. The EP composite with only 5% PZS@ZIF-67 passed the UL-94 test with the V-0 rating, and the LOI value improved from 25.5 to 31.9% compared to that of the pristine EP. Moreover, the epoxy composite containing PZS@ZIF-67 presented lower pHRR, THR, TSP, and FIGRA values compared to those of the sample containing the same amount of unmodified PZS. The enhanced fire safety performance of PZS@ZIF-67 was obtained by engineering ZIF-67 on the PZS microspheres. The preponderances of PZS@ZIF-67 for the comprehensive performance included an improved thermo-oxidative stability and yield of the char residues, multi-elements synergistic flame-retardant effect, and effective dispersion effectiveness that allows an enhanced interface interaction within the epoxy matrix. The fine dispersion state and rough surface structure of the flame-retardant fillers enhance the composites’ mechanical properties at relatively low content. The core–shell PZS@ZIF-67 material fabricated successfully in this study may play a critical role in improving fire safety performance and many other fields due to its versatility.

## Figures and Tables

**Figure 1 polymers-13-02646-f001:**
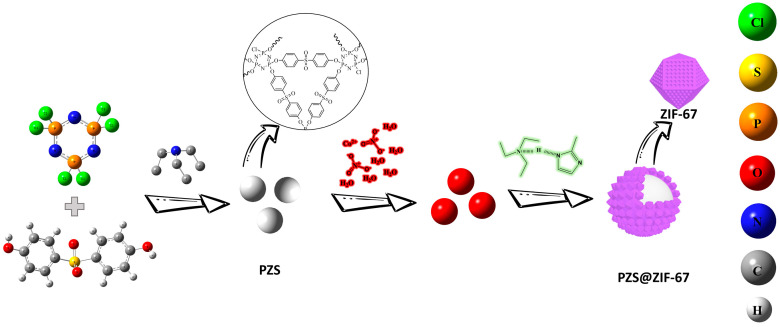
Synthetic diagram describing the preparation of PZS@ZIF-67.

**Figure 2 polymers-13-02646-f002:**
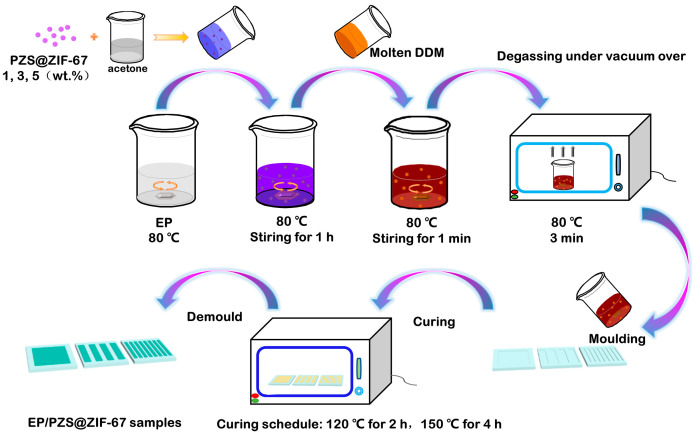
Schematic of the preparation of the EP/PZS@ZIF-67 composite.

**Figure 3 polymers-13-02646-f003:**
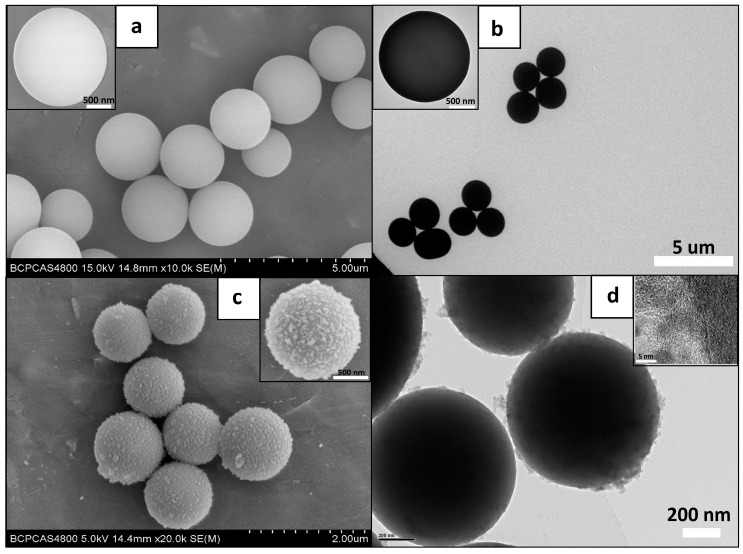
SEM and TEM images of PZS (**a**,**b**) and PZS@ZIF-67 (**c**,**d**).

**Figure 4 polymers-13-02646-f004:**
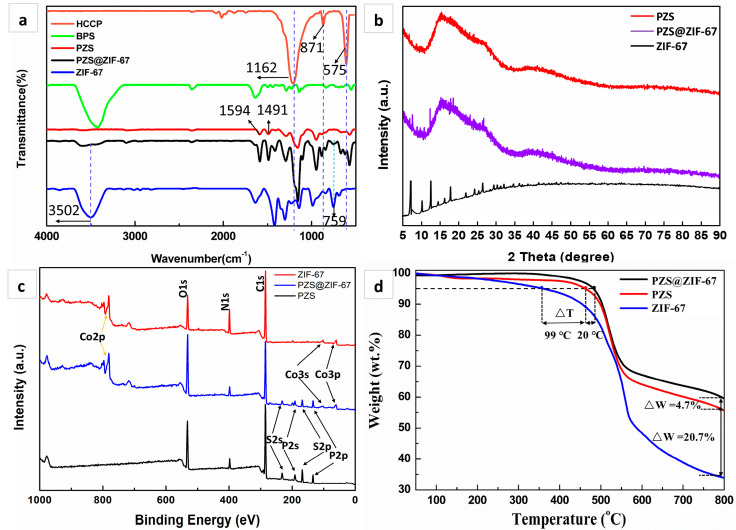
(**a**) FTIR spectra; (**b**) Wide-angle XRD patterns; (**c**) XPS survey spectra; (**d**) TGA curves of ZIF-67, PZS, and PZS@ZIF-67.

**Figure 5 polymers-13-02646-f005:**
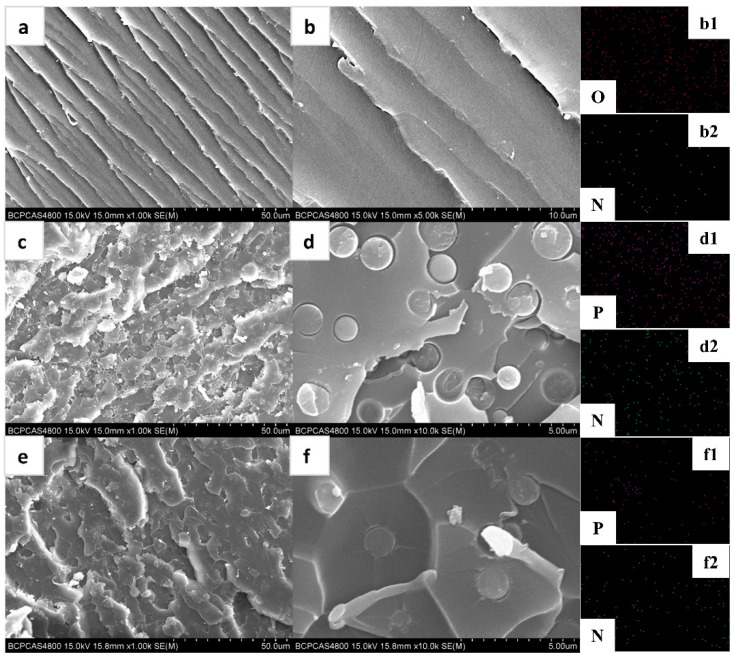
SEM and magnified SEM images of (**a**,**b**) EP, (**c**,**d**) EP/5% PZS, and (**e**,**f**) EP/5% PZS@ZIF-67; (**b1**,**b2**) element mapping of EP; (**d1**,**d2)** element mapping of EP/5% PZS; (**f1**,**f2**) element mapping of EP/5% PZS@ZIF-67.

**Figure 6 polymers-13-02646-f006:**
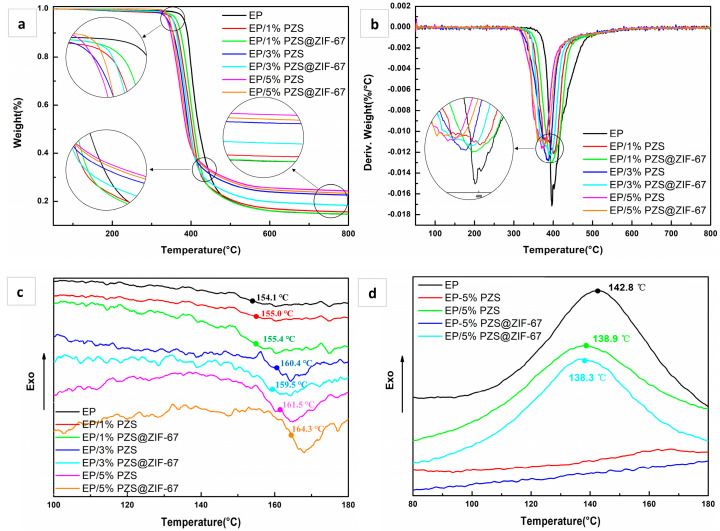
TGA (**a**), DTG (**b**), DSC (**c**), and DSC during curing process; (**d**) curves of pristine EP and its composites.

**Figure 7 polymers-13-02646-f007:**
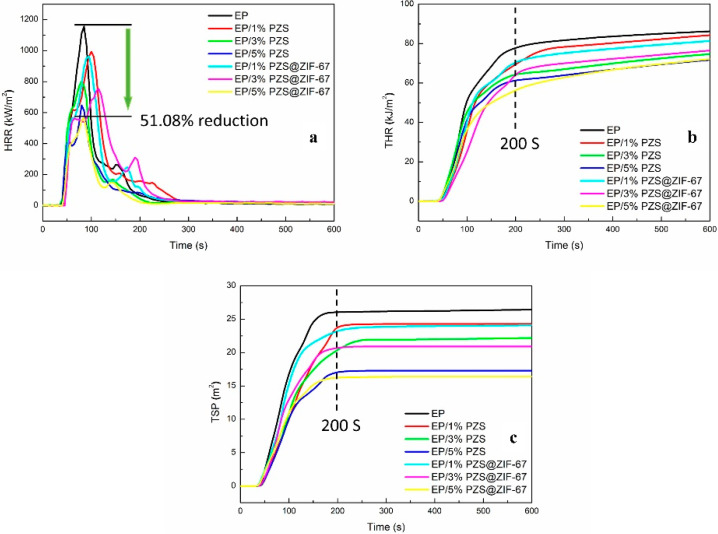
HRR (**a**), THR (**b**), and TSP (**c**) vs. time curves of pristine EP and its composites.

**Figure 8 polymers-13-02646-f008:**
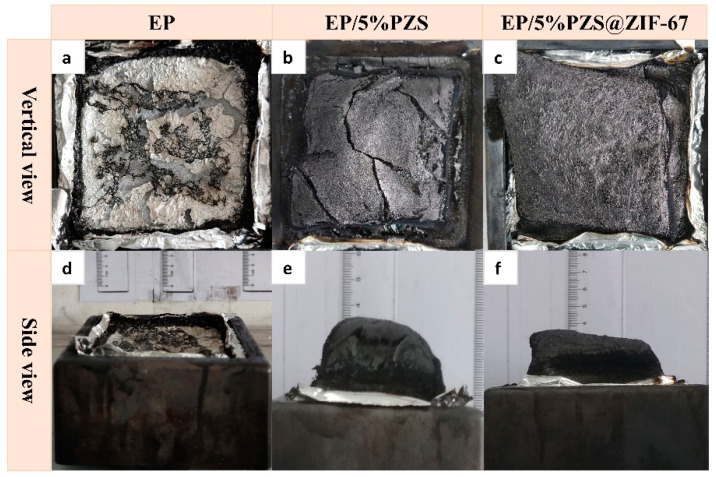
(**a**) Vertical view and (**d**) front view of EP char residue after the cone calorimeter test, (**b**) Vertical view and (**e**) front view of EP/5% PZS char residue after the cone calorimeter test, (**c**) Vertical view and (**f**) front view of EP/5% PZS@ZIF-67 char residue after the cone calorimeter test, respectively.

**Figure 9 polymers-13-02646-f009:**
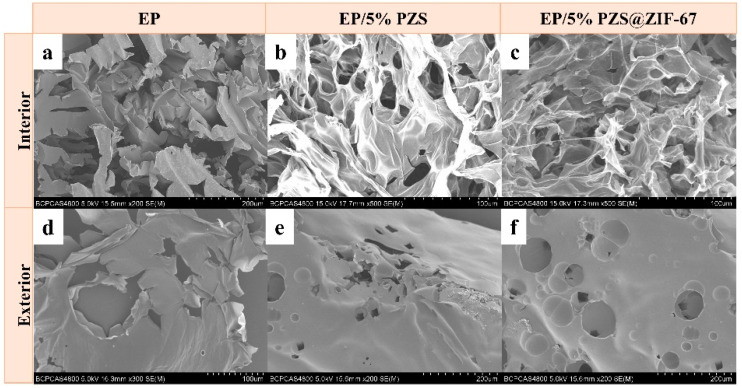
(**a**,**d**) SEM images of char from EP, (**b**,**e**) SEM images of interior and exterior char from EP/5% PZS, (**c**,**f**) SEM images of interior and exterior char from EP/5% PZS@ZIF-67, respectively.

**Figure 10 polymers-13-02646-f010:**
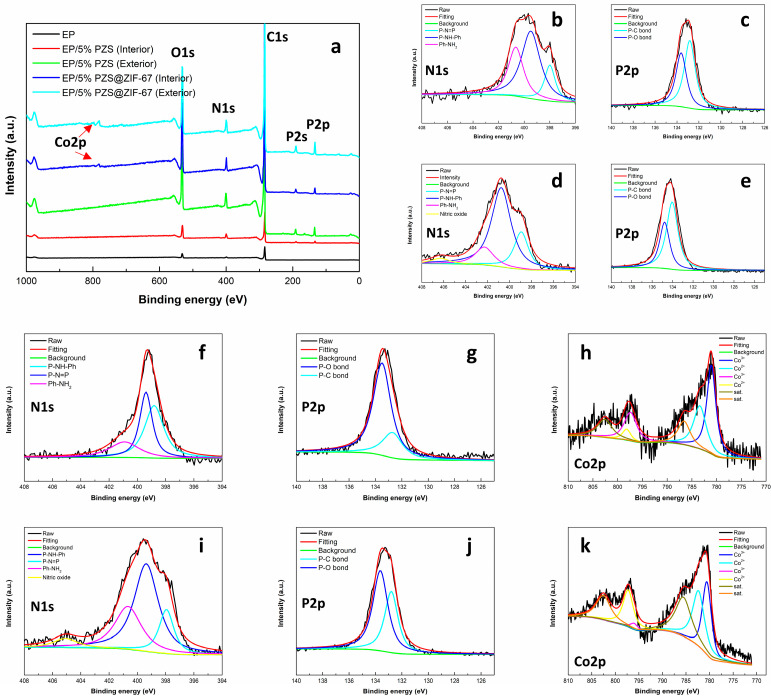
(**a**) XPS full spectra of char residue of EP, EP/5% PZS, and EP/5% PZS@ZIF-67; N1s and P2p of (**b**,**c**) interior and (**d**,**e**) exterior char from EP/5% PZS, N1s, P2p, and Co2p of (**f**–**h**) interior and (**i**–**k**) exterior char from EP/5% PZS@ZIF-67.

**Figure 11 polymers-13-02646-f011:**
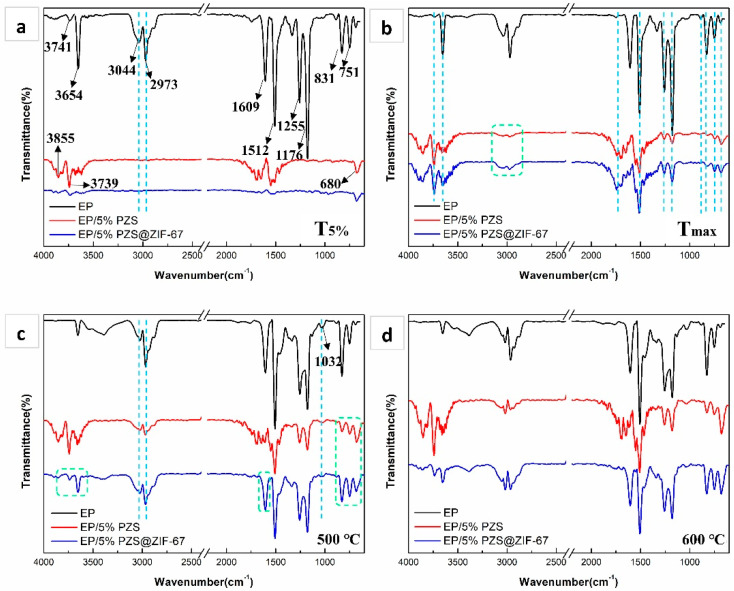
FTIR spectra of the gaseous products of the EP, EP/5% PZS, and EP/5% PZS@ZIF-67 at T_5%_ (**a**), T_max_ (**b**), 500 °C (**c**), and 600 °C (**d**).

**Figure 12 polymers-13-02646-f012:**
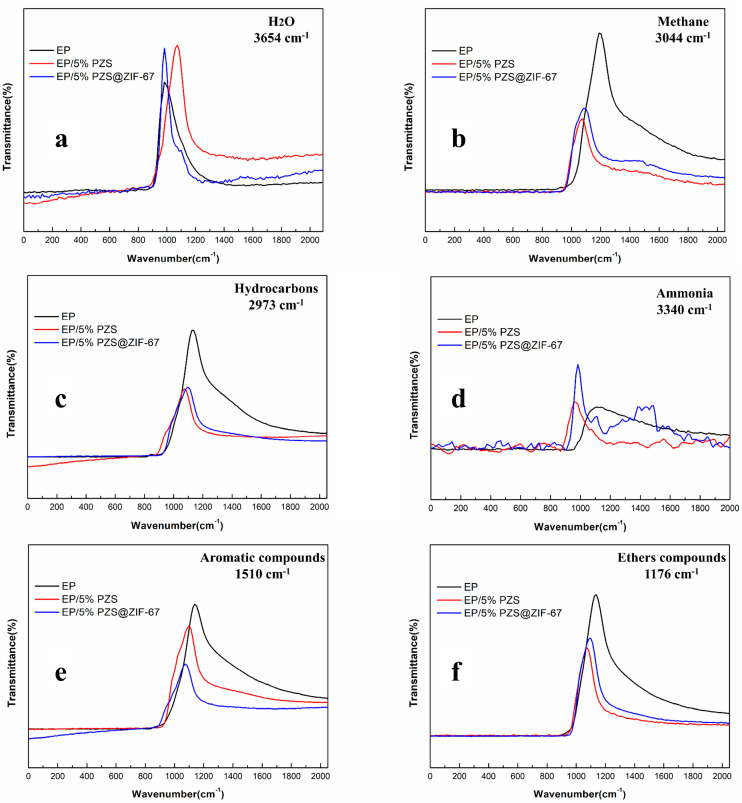
FTIR spectra of pyrolysis products of EP, EP/5% PZS, and EP/5% PZS@ZIF-67 vs. time: (**a**) H_2_O; (**b**) methane; (**c**) hydrocarbons; (**d**) ammonia; (**e**) aromatic compounds; and (**f**) ether compounds.

**Figure 13 polymers-13-02646-f013:**
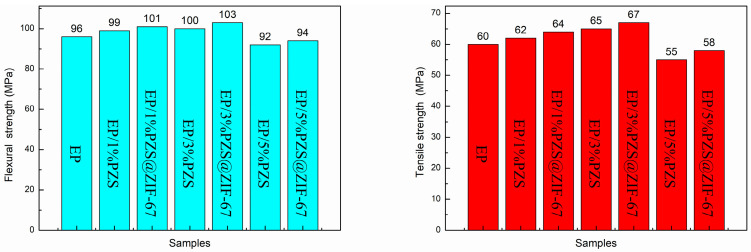
Flexural and tensile strength of EP and its composites.

**Figure 14 polymers-13-02646-f014:**
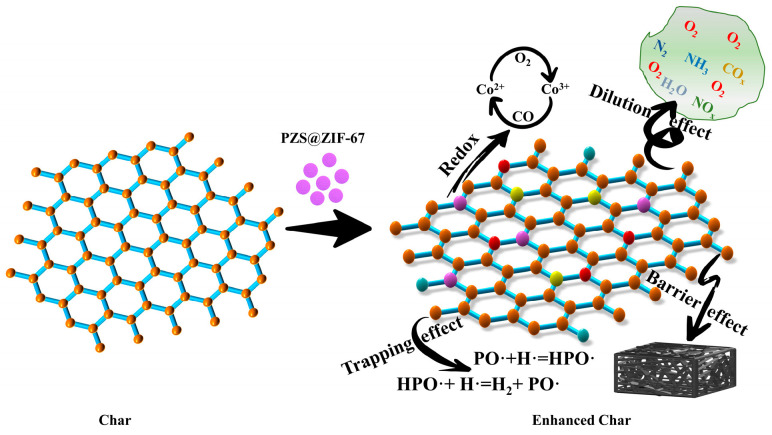
The possible flame-retardant mechanism of PZS@ZIF-67 composite in epoxy resin.

**Table 1 polymers-13-02646-t001:** TGA data of ZIF-67, PZS, and PZS@ZIF-67.

Sample	T_5%_ (°C)	T_max_ (°C)	Residues at 800 °C (%)
PZS@ZIF-67	483	517	60.2
PZS	463	520	55.5
ZIF-67	364	510 & 558	34.8

**Table 2 polymers-13-02646-t002:** TGA detailed data of EP and its composites.

Sample	T_5%_ (°C)	T_max_ (°C)	Residues at 600 °C (%)	Residues at 800 °C (%)
Pristine EP	380.7	396.2	16.18	14.60
EP/1% PZS	351.7	394.8	17.09	15.64
EP/1% PZS@ZIF 67	361.0	395.5	16.80	14.90
EP/3% PZS	338.3	385.8	23.91	22.56
EP/3% PZS@ZIF-67	352.3	392.7	19.82	18.35
EP/5% PZS	337.0	370.3	25.65	24.27
EP/5% PZS@ZIF-67	344.7	377.6	24.79	23.27

**Table 3 polymers-13-02646-t003:** The feed ratio of each component in curing DSC test.

Sample	Epoxy Resin	DDM	PZS	PZS@ZIF-67
Pristine EP	60	15	-	-
EP-5% PZS	60	0	3.16	-
EP-5% PZS@ZIF-67	60	0	-	3.16
EP/5% PZS	60	15	3.95	-
EP/5% PZS@ZIF-67	60	15	-	3.95

**Table 4 polymers-13-02646-t004:** LOI and UL-94 test results of EP and its composites.

Sample	LOI (%)	UL-94	
		Rating	Dripping
Pristine EP	25.5	No rating	Yes
EP/1% PZS	28.8	No rating	Yes
EP/1% PZS@ZIF-67	30.1	No rating	Yes
EP/3% PZS	30.5	No rating	No
EP/3% PZS@ZIF-67	31.2	V-1	No
EP/5% PZS	31.5	V-0	No
EP/5% PZS@ZIF-67	31.9	V-0	No

**Table 5 polymers-13-02646-t005:** Cone calorimeter data of pristine EP and its composites.

	Parameter	pHRR (kW/m^2^)	THR (MJ/m^2^) at 200 s	TSP (m^2^) at 200 s	Char Residue (%)	FIGRA (kW m^−2^ s^−1^)
Sample						
Pristine EP		1156.16	78.16	26.01	2.83	13.72
EP/1% PZS		998.45	69.72	23.87	9.69	10.08
EP/1% PZS@ZIF-67		957.56	70.53	23.28	11.77	10.05
EP/3% PZS		799.13	65.34	20.43	13.59	9.87
EP/3% PZS@ZIF-67		748.09	64.50	20.67	14.21	6.70
EP/5% PZS		640.69	61.29	17.11	15.95	7.92
EP/5% PZS@ZIF-67		565.59	56.07	16.16	16.74	6.72

**Table 6 polymers-13-02646-t006:** XPS data of the residual chars of Pristine EP, EP/5% PZS, and EP/5% PZS@ZIF-67.

Signal	Binding Energy (eV)	Area					
		Pristine EP		EP/5% PZS		EP/5%PZS@ZIF-67	
		Exterior Char	Interior Char	Exterior Char	Interior Char	Exterior Char	Interior Char
C 1s (C-H,C-C)	284.6	87.66		82.1	84.09	79.24	81.68
C 1s (C-O)	286.8						
C 1s (C=O)	288.4						
O 1s		7.97		9.19	8.15	11.77	11.37
P 2p		-		5.32	2.80	4.23	2.68
N 1s		4.38		3.39	4.96	4.09	3.97
Co 2p		-		-	-	0.66	0.3

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
