# Peer review of "Synthesis of a Reactive Template-Induced Core–Shell PZS@ZIF-67 Composite Microspheres and Its Application in Epoxy Composites"

_polymers, 2021, doi:10.3390/polym13162646_

Round 1

Reviewer 1 Report

“line 28: the superior dispersibility and more vital interaction of fillers in the epoxy matrix, which ameliorates the char layer’s graphitization and exerts a synergistic flame retardant effect on epoxy composites, compared with unmodified epoxy resin.” how to verify it ?

Author Response

Responds to the reviewer’s comments:

For reviewer #1:

We are grateful to reviewer #1 for his/her effort reviewing our paper and his/her positive feedback. The summary of our work as written by this reviewer is precise. Here below we address the questions and suggestions raised by the reviewer #1.

Thank you very much for your useful comments and suggestions on our manuscript, which help us a lot to improve our paper. We have made careful revisions on the original manuscript according to the comments. We hope the revised edition will meet the Polymers journal’s standard for publication.

Comment 1:

“line 28: the superior dispersibility and more vital interaction of fillers in the epoxy matrix, which ameliorates the char layer’s graphitization and exerts a synergistic flame retardant effect on epoxy composites, compared with unmodified epoxy resin.” how to verify it ?

Respond 1:

      Thank you for your careful reading of our manuscript. After careful discussion of your suggestion, we decided to delete this sentence. Whether the dispersion of filler in epoxy matrix can promote the graphitization of residual carbon remains to be further discussed. This statement is indeed not verified in subsequent work. Thank you very much for your suggestions, which make our manuscript more rigorous.

Your Suggestions are of great significance to my future work. Special thanks to you for your good comments.

Reviewer 2 Report

Comments are in the attachment.

Author Response

Responds to the reviewer’s comments:

For reviewer #2:

We are grateful to reviewer #2 for his/her effort reviewing our paper and his/her positive feedback. The summary of our work as written by this reviewer is precise. Here below we address the questions and suggestions raised by the reviewer #2.
We have carefully addressed all the reviewer's concerns. Please below our replies. We hope he/she is satisfied with our answers and the new data we provided. Changes highlighted in yellow have been made accordingly, in the revised manuscript and in the revised supplementary accordingly, in the revised manuscript and in the revised supplementary information.

Comment 1:
In which industry will these coatings be used? This question is very important, since it allows us to understand what physicomechanical, chemical or mechanical properties should be investigated.
Respond 1:
Thank you for your careful reading of our manuscript. The flame retardant PZS@ZIF-67 has uniform structure and excellent thermal stability. It can be used not only for flame retardant of polymer materials, but also for toughening of phenolic resin, epoxy resin and other materials. However, the particle size of PZS@ZIF-67 prepared by us is about 1.5 um. If we continue to explore the toughening effect, the particle size of polaphosphazene microspheres (PZS) can be controlled below 500 nm. 
At the same time, the research of polyphosphazene microspheres (PZS) for coating has also been reported, see the paper "Li AiYuan. Research Progress in the Application of Polyphosphazene in Coatings. Monograph on special Paint and Coatings, 2007, 10(9), 48-51 ". Your opinion is of practical significance. In addition, we are considering whether it can be used in the study of polyphosphazene elastomers

Comment 2:
The fonts of the Figs 2, 4, 8-10 are very small. This complicates the analysis of the results; it needs to be increased.
Respond 2:
Thank you for your instructive suggestions. I have enlarged the images appropriately for your review and highlighted in yellow it accordingly. In addition, for Figure 8 (now Figure 10), there is too much information, we did not modify the image for aesthetic purposes. If you think it is necessary to modify, we will do further typesetting

Comment 3:
Schemes 1-3 are also figures. It must be included in the general numbering of figures.
Respond 3:
Thank you for your careful reading of our manuscript. We are very sorry for our incorrect marking. Schemes 1-3 have been included in the general numbering of figures and the changes have been marked.

Comment 4:
I propose to supplement the introductory part of the article with a review of the articles:
Stukhlyak, P.D., Mytnyk, M.M. & Orlov, V.O. Influence of Boundary Interlayers on Properties of Composite Polymeric Materials (a Review). Materials Science 37, 80–86 (2001). https://doi.org/10.1023/A:1012338422984
A. Buketov, P. Maruschak, O. Sapronov, M. Brailo, O. Leshchenko, L. Bencheikh & A. Menou (2016) Investigation of thermophysical properties of epoxy nanocomposites, Molecular Crystals and Liquid Crystals, 628:1, 167-179, https://doi.org/10.1080/15421406.2015.1137122
Buketov, A., Maruschak, P., Sapronov, O., Zinchenko, D., Yatsyuk, V., & Panin, S. (2016). Enhancing performance characteristics of equipment of sea and river transport by using epoxy composites. Transport, 31(3), 333–342. https://doi.org/10.3846/16484142.2016.1212267
Respond 4:
Thank you for your valuable and thoughtful comments. I further learned about nano-flame retardant and epoxy composite application research by reading these articles you recommended. These articles have very good references for our manuscripts. Moreover, we have cited them in the relevant paragraphs of our manuscripts. 
In page 1 line 36, “Investigation of thermophysical properties of epoxy nanocomposites, Molecular Crystals and Liquid Crystals, 628:1, 167-179” was added; 
In page 1 line 36, “Enhancing performance characteristics of equipment of sea and river transport by using epoxy composites. Transport, 31(3), 333–342” were added;
In page 7 line 250, “Influence of Boundary Interlayers on Properties of Composite Polymeric Materials (a Review). Materials Science 37, 80–86 (2001)”, and “Polymer Degradation and Stability 94 (2009) 291-296” were added;
We have carefully read the article recommended by you, so that I have a deeper understanding of the impact of interfacial interlayer on the properties of composites, which will play a good role in guiding my subsequent work. According to your helpful advice, we have enriched the recent research progress in the introduction. Thank you again.

Comment 5:
I propose to show the dispersion (scutter) of mechanical properties in Fig. 11. This will make it possible to assess not only their properties, but also their stability in a sample of samples.
Respond 5:
Thank you for your valuable and thoughtful comments. By observing the spline section after mechanical test, it is significant to explore the stability of filler in the matrix. At present, the obstacle is that I have graduated and left school, and it is very difficult to carry out the supplementary test in the experiment, which is exactly what I worry about. The dispersion test of flame retardant filler mentioned above is used for brittle fracture of spline in liquid nitrogen, which also has certain reference value. Your suggestions are of guiding significance for improving the systematization of our manuscript. In the future work, I will systematically examine the possible problems and explore them.

Comment 6:
On the basis of what standards were mechanical, fractographic and physicochemical tests carried out. This information needs to be added to the text and to the list of links.
Respond 6:
Thank you for your valuable comments. The test criteria for mechanical properties of composites have been added to the manuscript, which will further improve our manuscript. 
In page 4 line 170, “The mechanical properties of splines were tested by the general testing machine (CMT-4104, MTS Systems (China) Co.,Ltd.) in accordance with GB/T 2567-2008 "Ten-sile Test and Bending Test Method".” was added;

Comment 7:
The scientific value of the "black rectangles" b1, b2, d1, d2, f1, f2 is shown in Fig. 3 requires additional justification. 
Respond 7: 
"Black rectangles: b1, b2, d1, d2, f1, f2" is an additional energy spectrum analysis test in the SEM test to observe the distribution of elements. However, because the detection limit is not high, it is difficult to detect the small content of cobalt element. From this test we want to show that the flame retardant is distributed evenly in the matrix. Therefore, we can only show the dispersity of flame retardant in epoxy by phosphorus element

Comment 8:
From the data in Fig. 3 that the distribution of the filler in the epoxy matrix is fairly uniform. At the same time, it is seen that the formation of outer polymer layers is possible near the particles. Their presence can affect the mechanical strength and stability of the properties of the epoxy composite.
Respond 8: 
Thank you for your valuable advice. On the whole, the properties of a composite depend on properties of the matrix, filler, and boundary interlayers. Boundary interlayers have a considerable influence on the properties of polymeric composite materials. It's true that I haven't explored the boundary conditions enough. The introduction of active fillers that interact with the polymer and, simultaneously, are centers of structure formation increases the defective and structural inhomogeneity of materials. In the following work, I should further understand the impact of the introduction of fillers and the increase of their content on other physical and chemical properties of composites. At the same time, the article you recommended to me made a systematic elaboration on the knowledge of boundary interlayer, which enabled me to further understand its phenomenon and mechanism.

Your Suggestions are of great significance to my future work. Special thanks to you for your good comments.

Reviewer 3 Report

The manuscript entitled "Synthesis of a Reactive Template-Induced Core–Shell 2PZS@ZIF-67 Composite Microspheres and Its Application in 3
Epoxy Composites" by Kunpeng Song et al. is of high scientific and practical interest. The presented manuscript is interesting as well as highly relevant to the Journal readers. The manuscript topic is within the scope of the polymer Journal. Moreover, the manuscript is well written. I advise the acceptance of the present manuscript 

Author Response

Responds to the reviewer’s comments:

For reviewer #3:

We are grateful to reviewer #3 for his/her effort reviewing our paper and his/her positive feedback. The summary of our work as written by this reviewer is precise.

Your Suggestions are of great significance to my future work. Special thanks to you for your good comments.

Round 2

Reviewer 2 Report

Accept.